# Effect of High Strain Rate on Adiabatic Shearing of α+β Dual-Phase Ti Alloy

**DOI:** 10.3390/ma14082044

**Published:** 2021-04-19

**Authors:** Fang Hao, Yuxuan Du, Peixuan Li, Youchuan Mao, Deye Lin, Jun Wang, Xingyu Gao, Kaixuan Wang, Xianghong Liu, Haifeng Song, Yong Feng, Jinshan Li, William Yi Wang

**Affiliations:** 1State Key Laboratory of Solidification Processing, Northwestern Polytechnical University, Xi’an 710072, China; haofang85@163.com (F.H.); peixuan_li@mail.nwpu.edu.cn (P.L.); myc2525@163.com (Y.M.); nwpuwj@nwpu.edu.cn (J.W.); yfeng@c-wst.com (Y.F.); ljsh@nwpu.edu.cn (J.L.); 2Western Superconducting Technologies Co., Ltd., Xi’an 710018, China; kingsin@c-wst.com (K.W.); xhliu@c-wst.com (X.L.); 3CAEP Software Center for High Performance Numerical Simulation, Institute of Applied Physics and Computational Mathematics, Beijing 100088, China; lin_deye@iapcm.ac.cn; 4Laboratory of Computational Physics, Institute of Applied Physics and Computational Mathematics, Beijing 100088, China; gao_xingyu@iapcm.ac.cn (X.G.); song_haifeng@iapcm.ac.cn (H.S.)

**Keywords:** deformation and fracture, microstructure, adiabatic shear bands, Schmid factor, dual-phase

## Abstract

In the present work, the localized features of adiabatic shear bands (ASBs) of our recently designed damage tolerance α+β dual-phase Ti alloy are investigated by the integration of electron backscattering diffraction and experimental and theoretical Schmid factor analysis. At the strain rate of 1.8 × 10^4^ s^−1^ induced by a split Hopkinson pressure bar, the shear stress reaches a maximum of 1951 MPa with the shear strain of 1.27. It is found that the α+β dual-phase colony structures mediate the extensive plastic deformations along α/β phase boundaries, contributing to the formations of ASBs, microvoids, and cracks, and resulting in stable and unstable softening behaviors. Moreover, the dynamic recrystallization yields the dispersion of a great amount of fine α grains along the shearing paths and in the ASBs, promoting the softening and shear localization. On the contrary, low-angle grain boundaries present good resistance to the formation of cracks and the thermal softening, while the non-basal slipping dramatically contributes to the strain hardening, supporting the promising approaches to fabricate the advanced damage tolerance dual-phase Ti alloy.

## 1. Introduction

Titanium (Ti) alloys are widely used in aerospace and biomedicine, attributed to their excellent high specific strength, good corrosion resistance, and high heat resistance [1,2,3]. In the development of advanced damage-tolerant Ti alloys, it is a challenge to dramatically enhance their mechanical properties under extreme conditions, such as high temperature and pressure, severe corrosion environments, or high loading rates. Recently, defect engineering has been considered as an effective strategy for modifying the local microstructures, properties, and performance of advanced metal materials [2,4,5,6,7,8]. Without changing the chemical composition, the introduction of defects provides more freedom to optimize microstructures and mechanical properties [4,5,8]. In particular, dual-phase materials [9,10,11] present ultra-strong and ductile behaviors through solid solution strengthening, the grain refinement effect, and precipitation hardening, which excellently deal with the planar defects, including stacking faults, grain boundaries, and phase boundaries. Through combining a high volume fraction of pyramidally arranged non-shearable super-refined α precipitates in the constrained β matrix, the ultimate strength of α+β dual-phase Ti-15Mo-3Nb-2.7Al-0.2Si (wt %) alloy (the β-21S or TB8 alloy) can be optimized in the range of 1–2 GPa [12]. Similarly, the α+β dual-phase Ti-3Mo-3Cr-2Fe-2Al alloy presents an excellent combination of ultimate tensile strength and ductility at 1324 MPa and 0.37, respectively [13]. Attributed to the twinning induced plasticity (TWIP) and transformation-induced plasticity (TRIP) effects, the strain-transformable Ti alloys display a superior combination of strength, ductility, and strain-hardening [14]. The sliding of α/β interfacial and colony boundaries plays the dominant role in the deformation of a newly developed Ti-0.85Al-4V-0.25Fe-0.25Si-0.15O alloy [15]. Moreover, in hexagonal close-packed (HCP) structures, the selection of key slip systems is generally controlled by the stacking fault energies in the basal and the prismatic planes, which dominate the slip process to form either twins or dislocations [2,7,16]. The Schmid law [1,16,17,18] provides an efficient approach to estimate the most possible activated slip variants for the given orientation relationships, the tensile direction and the crystallographic orientations, and thus to estimate the corresponding plastic deformation behaviors.

The strain-rate-dependent mechanical properties and structural evolutions should be revealed comprehensively [19,20,21,22,23]. Adiabatic shear bands (ASBs) are obtained at a high strain rate, yielding from the competition between thermal softening and strain/stress-rate hardening within the narrow planar region [20,24,25]. Although twinning is the most dominant plastic deformation behavior for HCP structures at ambient temperature, it is suppressed by dislocation gliding at high temperatures, which is for colony microstructures [17,26,27]. For instance, in the commercial α+β dual-phase Ti-10V-2Fe-3Al alloy with TWIP/TRIP properties, the strain-induced martensite α” is treated as a relaxation mechanism at the α/β interface [14]. With the enhancement in strain rate from 500 to 1000 s^−1^, the stress-induced martensitic transformation was observed in the TB8 alloy [28]. At the strain rate of 1000 s^−1^, multiple deformation mechanisms were comprehensively investigated in Ti-25Nb-3Zr-3Mo-2Sn, presenting the dynamic deformation sequences as {332}<113> and {112}<111> mechanical twining + stress-induced α” and ω phase transformations + dislocation slip at 293K → {332}<113> and {112}<111> mechanical twining + dislocation slip at 573K → dislocation slip only at 873 K [29]. The softening behavior of dual-phase Ti17 alloy (Ti-5Al-4Cr-4Mo-2Sn-2Zr) at a high temperature is attributed to the combinations of dynamic recrystallization, dynamic transformation, adiabatic heating, and morphological texture evolution [30]. Instead of thermal softening mechanisms, the microstructure evolutions or transformations play an important role in the initiation of ASB [21,24,31,32,33]. It is essential to reveal the microstructure–property relationship under dynamic loading in order to develop advanced damage-tolerant Ti alloys. Therefore, the α+β dual-phase Ti alloy (Ti-6Al-2Cr-2Mo-2Nb-2Sn-2Zr) fabricated by Western Superconducting Technologies Co, Ltd (Xi’an, China) was utilized to comprehensively reveal the localized features of ASBs in the present work.

## 2. Materials and Methods

In the present work, the hat-shaped specimen [34] of the annealed forging α+β Ti dual-phase alloy was deformed at ambient temperature using split Hopkinson pressure bar (SHPB), which conventionally captures the well-controlled and designed localized shear in the study of large strain and high strain rate deformation. In order to show earlier behaviors of shear localization, a lower shear strain rate of 1.8 × 10^4^ s^−1^ was deliberately selected. Three samples were utilized in the SHPB tests, but the one yielding the best results is reported and was analyzed further. Afterward, the standard metallographic procedure was utilized to prepare all samples, which were etched in a solution of 8% tetrafluoroboric acid for 90 s for the final characterizations. For the reference state, the static tensile properties were tested at the strain rate of 0.01 based on the national standard of GB/T 228.1-2010. Electron backscattered diffraction (EBSD) analysis was executed on a JEOL 7800F field emission SEM equipped with SymmetryTM (Oxford Instruments, Oxfordshire, UK). The accelerating voltage was 20 kV, the probe current was 14 nA, the scanning speed was 815 Hz, and the scanning step sizes of 0.15 and 1 μm were used for areas of 0.16 and 2.88 mm^2^, respectively.

In line with the Schmid law (τ = σ cosφ cosλ), the slip system, with either a maximum position or minimum negative SF (m = cosφ cosλ), was activated by the largest shear stress (τ). Here, λ is the angle between the slip direction and external force (σ) and φ is the angle between the normal of the slip plane and center axis. Based on the geometry of the hat-shaped specimen [34], the slip plane was fixed by the weakest regions, as presented by a group of planes paralleling to the designed shear region. Therefore, the global <phi> depended on the sample’s geometry structure, and cosφ was fixed at $1/\sqrt{5}$.

## 3. Results and Discussion

### 3.1. Microstructure Characterizations

Based on the EBSD analysis of the as-received specimen in Figure 1, we found that the α+β dual-phase Ti alloy consisted of equiaxed α_p_ with an average size of about 10 μm and lamellar secondary α phases (α_s_) with a thickness of about 0.9 μm. The amounts of α and β phases were 80.3% and 18.9%, respectively, which are identified in red and blue, respectively. The high-angle grain boundaries among the colony structures of the as-received specimen were constructed by α and β phases. The amount of these equiaxed α_p_ grains were about 85% without forming texture. 

### 3.2. Mechanical Response at High Strain Rate

Based on the geometry of the hat-shaped specimen, the force applied to the shear region of the hat-shaped specimen in the SHPB test was obtained from the strain gauges on the incident and transmitted bars. The following equations [35] were used to calculate the shear stress *τ_s_*, shear strain *γ_s_*, and shear strain rate $\dot{\gamma_{s}}$ within the shear region:(1)$$\tau_{s}\left(t\right)=\frac{E_{0}A_{b}}{A_{s}}\varepsilon_{t}$$
(2)$$\gamma_{s}=-\frac{C_{0}}{d_{1}-d_{0}}\int_{0}^{t}\left(\varepsilon_{i}-\varepsilon_{t}\right)dt$$
(3)$$\dot{\gamma_{s}}=\frac{C_{0}}{d_{1}-d_{0}}\left(\varepsilon_{i}-\varepsilon_{t}\right)$$
where *d*_0_ and *d*_1_ are the outside diameter of the hat and the inner diameter of the brim ring of specimen, respectively; *C*_0_ and *E*_0_ are the elastic wave speed and elastic modulus of Hopkinson bars, respectively; *A_s_* and *A_b_* are the area of shear section in the hat-shaped specimen and the area of cross-section in the transmitted bar, respectively; and *ε_i_* and *ε_t_* are the elastic strains of incident and transmitted bars, respectively.

At the strain rate of 1.8 × 10^4^ s^−1^, this alloy presented an excellent damage tolerance capability, which had a maximum shear stress as high as 2 GPa at the strain-hardening stage, as shown in Figure 2. It can also be seen that this stress–strain curve consisted of four stages, including an elastic region, strain hardening, stable softening, and unstable softening. In particular, the shear stress increased with increased shear strain, and reached a maximum as high as 1951 MPa with a shear strain of 1.27, presenting an ultra-strong behavior compared with the classical high-strength Ti alloys. In the range of shear strain from 1.27 to 1.60, the shear stress slowly decreased by 104 MPa from the peak, attributed to the thermal softening during the adiabatic shear. The dynamic stored deformation energy until the work-hardening stage can be characterized as the integration of the stress–strain curve [36], which is much larger than those obtained from the static test. Correspondingly, it was expected that dynamic recrystallization would occur and result in the following softening. Moreover, the adiabatic shear behavior related to the microstructure evolution was revealed comprehensively, which is considered to be driven by the aforementioned stored energies.

### 3.3. Local Features of Adiabatic Shear Bands

In order to comprehensively reveal these stable and unstable softening mechanisms, multi-scale characterizations of the localized ASBs were essential, the typical features of which are displayed in Figure 3. The ASBs were mainly located at the α/β dual-phase colony structures, yielding severe plastic deformations along the α/β phase boundaries. This contributed to the formations of microvoids and cracks, thus resulting in stable and unstable softening behaviors. Moreover, these ASBs caused by the thermal softening and the stress concentrations were the preferred sites for nucleation, growth, and coalescence of microvoids, contributing to the formation of cracks in the ASBs after reaching the critical length [37]. Interestingly, these microvoids were generally located at the center of the ASBs, which are attributed to the gradients of temperature and stress from the center of the ASBs to the boundary of matrix [37]. Furthermore, there were several well-aligned parallel ASBs, constructing the primary and the subordinate shear paths/zones. Even within the primary shear path, it seemed as if these shear bands formed periodically were caused by the thermoplastic instability/softening, which highlights the significance of geometry and microstructures dominated by the shear angle, shear strain, and shear rate [38].

Figure 4 shows the full scans of the ASBs and the corresponding affected zones in terms of the phase map, the geometrically necessary dislocation (GND )map, and the coupling analysis of inverse pole figure and grain boundary map (IPF+GB). Different plastic strains between the left and the right sections may also yield different microstructure evolutions. In the phase maps, the estimated amounts of β phase decreased to less than 10%, which also indicates that the dominated shearing path was along the α/β phase boundaries, which caused the β grains within the elongated and the recrystallized extremely fine α grains to be indistinguishable. Dynamic recrystallization yielded a large amount of fine α grains dispersed along the shearing paths and in the ASBs, which could contribute to the formation of textures. It is understood that recrystallization is dominated by the entropic effect arising from the competition between the formation of dislocation and that of grain boundaries while the shear band instability emerges, which is attributed to the thermal heating occurring faster than heat dissipation [39]. Moreover, the thickness of the ASB-affected regions resulting in the collaborating plastic deformation was clearly captured by GND mapping. It seems as if the low-angle grain boundaries of the left side presented a good resistance to the formation of cracks and the thermal softening. The grain boundaries with high angles within the ASBs were also geometrical necessary boundaries, which is the same as previous observations in pure Ti [40]. In line with the grain boundary complexions [41,42,43], it is understood that free volumes exist at the boundaries, constructing the weakly bonded regions and contributing to the initiation of ASBs. Furthermore, the $\left\{0001\right\}\langle 11\bar{2}0\rangle$ slip systems highlighted in red in the IPF maps play an important role in the formation of ASBs, which is validated by the following theoretical analysis.

As displayed in Figure 5, the α grains were elongated while the β grains were difficult to capture in the colony structures. The solid and dashed arrows identify the cracks and the elongated α phase, respectively. It is understood that rotational dynamic recrystallization (RDR) yields a large amount of fine equiaxed α grains dispersed along the shearing paths and in ASBs, promoting the aforementioned stable softening and shear localization [20,36]. The equiaxed grains could be re-elongated by further deformation. In line with the progressive subgrain misorientation recrystallization, a mechanical subgrain rotation model to account for the recrystallized grains has been proposed and observed in ASBs in a number of materials [32,33,44]. The mechanical subgrain rotations at a high strain rate would assist the mechanism of recrystallizations. Recently, it was reported that the initiation of ASBs will occur ahead of the apparent temperature rise in Ti [23,24], motivating further investigation of the traditional well-accepted thermal-softening mechanism of ASBs. Thus, the local microstructure evolutions caused by shearing lead to dynamic recrystallizations and play an important role of softening during severe plastic deformation.

### 3.4. Discussion

During shock loading, uncommon slip systems might be activated. In general, plastic deformation initiates on the crystallographic plane with the highest Schmid factor when the stress resolved on that plane in the slip direction reaches a critical value. Based on the experimental analysis of Schmid factor, as shown in Figure 5 and Figure 6a,b, the basal and the prismatic slips mediated by $\left\{0001\right\}\langle 11\bar{2}0\rangle$ and $\left\{10\bar{1}0\right\}\langle 11\bar{2}0\rangle$ are the dominated slip systems. Therefore, the physical reasons for the selection of these two slip systems are essential, which can also verify whether the Schmid law breaks down. Under the condition of |cosλ(a,b,c)|≤1, the SF of a given slip direction <a,b,c> can be screened comprehensively. As displayed in Figure 6c,d, although the pyramidal slip systems have the largest SFs compared with those of basal and prismatic ones within the low angle ranges, they are difficult to activate because of the requirements of the largest critical resolved shear stress and the rotations of deformed grains. On the contrary, in the range of 25~35°, the basal and prismatic slips are the dominant approaches since both of them have the larger SFs, matching well with the present experimental observations. Since non-basal slipping enhances both the strength and ductility of an HCP structure [45], the $\left\{10\bar{1}0\right\}\langle 11\bar{2}0\rangle$ prismatic slips dramatically contribute to the aforementioned strain hardening. Finally, it is highlighted that the Schmid or the non-Schmid behaviors of the materials should be comprehensively investigated, revealing the variations in critical resolved shear stress and the selections of deformation model [18,26,46], providing the promising strategies to enhance damage tolerance properties.

## 4. Conclusions

In summary, the α+β dual-phase colony structures mediate the extensive plastic deformations along α/β phase boundaries, which contribute to the formations of ASBs, microvoids, and cracks, thus resulting in stable and unstable softening behaviors. Dynamic recrystallization yields a large amount of fine α grains dispersed along the shearing paths and in ASBs, promoting the softening and shear localization. On the contrary, the low-angle grain boundaries present a good resistance to the formation of cracks and thermal softening, while the non-basal slipping dramatically contributes to strain hardening, supporting the promising approaches to fabricating advanced damage-tolerant dual-phase Ti alloy.

## Figures and Tables

**Figure 1 materials-14-02044-f001:**
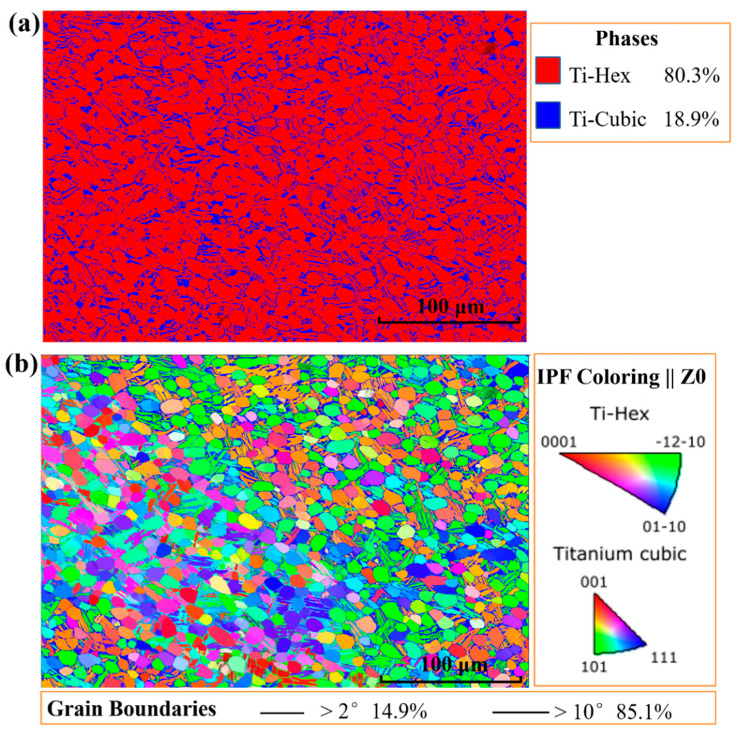
The electron backscattered diffraction (EBSD) analysis of the as-received dual-phase Ti alloy, (**a**) phase map; (**b**) the coupling analysis of inverse pole figure and the grain boundary map.

**Figure 2 materials-14-02044-f002:**
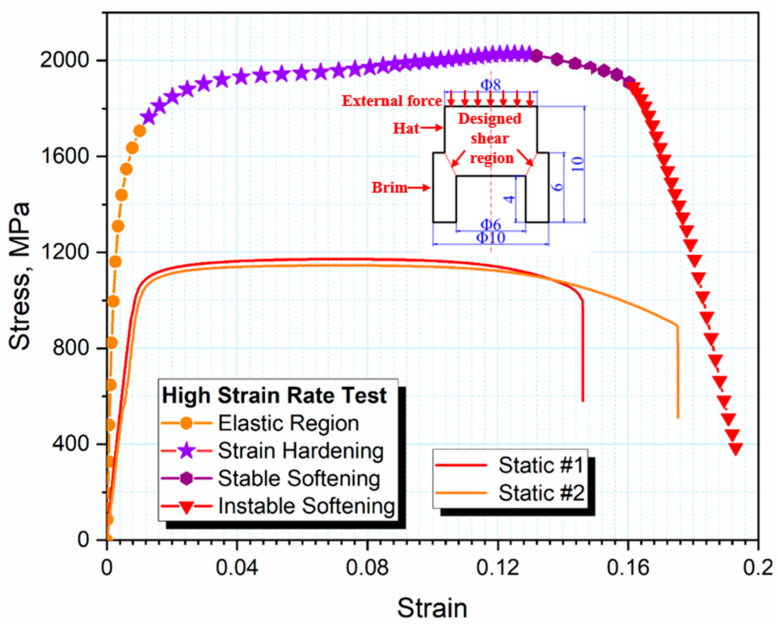
The stress–strain of dual-phase Ti alloy at the strain rate of 1.8 × 10^4^ s^−1^ and the static one (0.001 s^−1^) together with the geometry of the hat-shaped specimen.

**Figure 3 materials-14-02044-f003:**
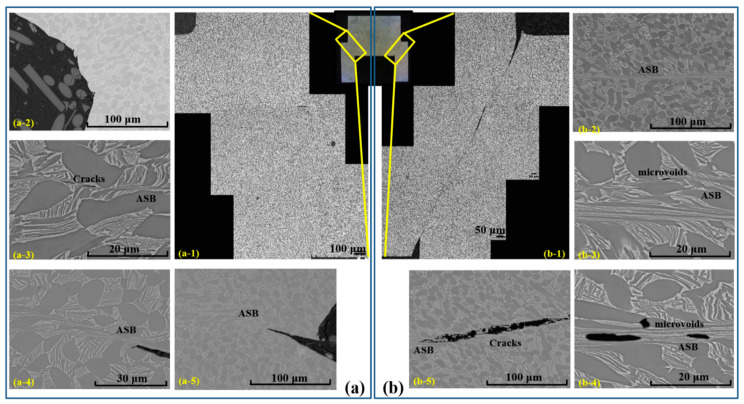
The multi-scale SEM characterizations of adiabatic shear bands in the hat-shaped specimen. The left part and the right part of the weak areas in the hat-shaped specimen are marked by yellow frames in (**a-1**) and (**b-1**), respectively. (**a-2**)~(**a-5**) reflect the typical features of the selected right region, such as ASB, microvoids and α+β phases, while the selected left region is identified in (**b-2**)~(**b-5**).

**Figure 4 materials-14-02044-f004:**
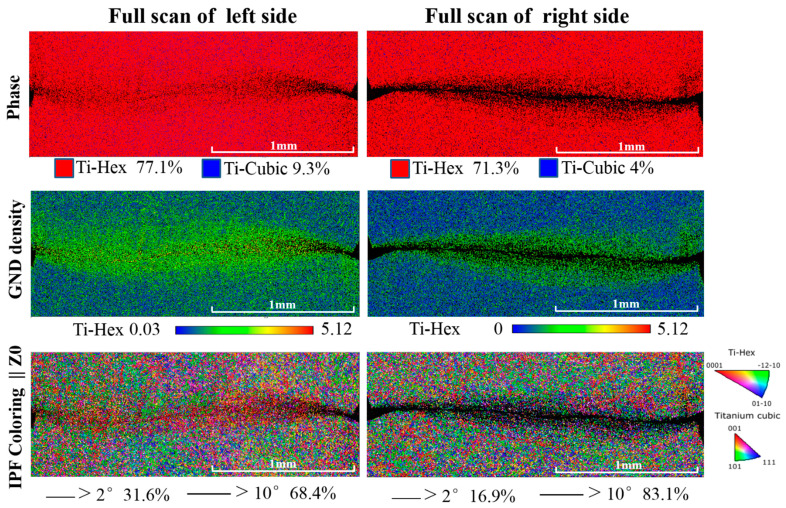
The EBSD features of the hat-shaped specimens in terms of phase map (Ph), geometrically necessary dislocation (GND) map, and the coupling analysis of inverse pole figure and grain boundary map (IPF+GB).

**Figure 5 materials-14-02044-f005:**
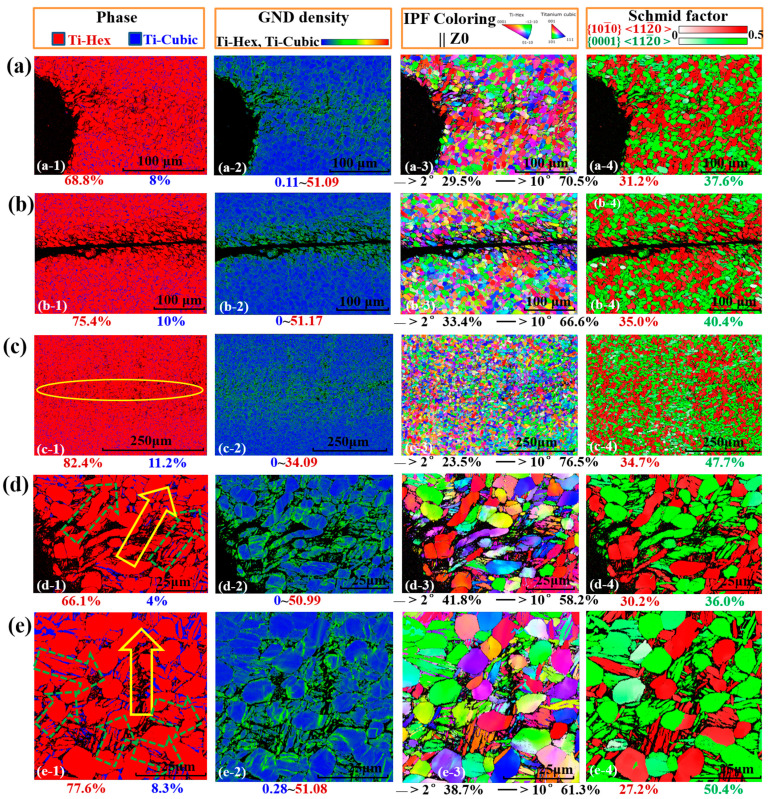
The typical features of ASBs and the affected zones in views of the phase map, GND map, and IPF+GB: (**a**–**c**) the cracks and ASB shear paths; (**d**,**e**) the localized shear deformation areas together with solid and dashed arrows identifying the cracks and the elongated α phase, respectively.

**Figure 6 materials-14-02044-f006:**
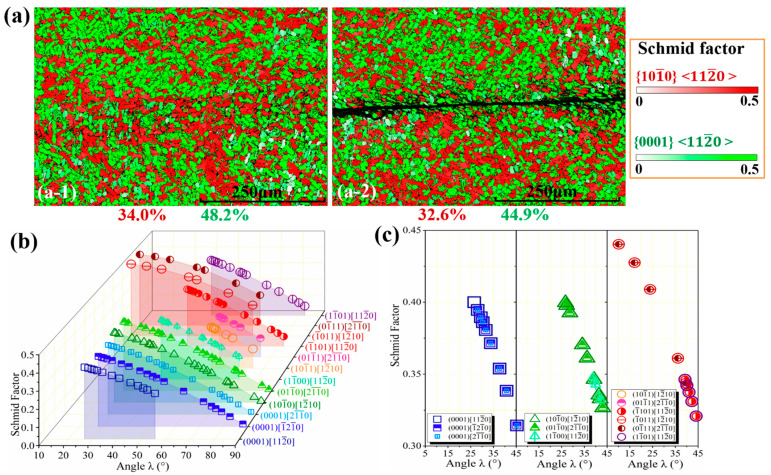
The theoretical Schmid factors (SFs) referring to experimental observations: (**a**) experimental characterizations of SFs of the left and the right side of the hat-shaped specimen; (**b**,**c**) 3D and 2D views of the theoretical SFs.

## Data Availability

Correspondence and requests for data and materials should be addressed to W.Y.W. (wywang@nwpu.edu.cn) and Y.X.D. (Eason@c-wst.com).

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
