# Peer review of "Effect of High Strain Rate on Adiabatic Shearing of α+β Dual-Phase Ti Alloy"

_materials, 2021, doi:10.3390/ma14082044_

Round 1

Reviewer 1 Report

Very nice paper. Only comment I would like to make is related to Fig2. How the Authors define elastic region? And possibly other regions? Why a part of the elastic region is non-linear?

Author Response

Please find the response letter in the attachment. Thanks again for the positive recommendation and constructive suggestions. 

Reviewer 2 Report

Experimental designed on damage tolerance α+β dual-phase Ti alloys were reported from the author as a futurable promis to enance the knowledge on this field of investigation by using electron backscattering diffraction and theoretical Schmid factor analysis. The deformation induced on the α+β dual-phase Ti alloy achieved a limit value and then  micro cracks and micro voids appears, only in some parts as showed from the figure. The article was well-written, the results match the discussion, and the literature is updated. The paper is suitable for publication; however, some minor point should be addressed before. An extese revision of english is required.

  • Abstract

Line 18 Definition “as hight as 1951MPa” is an expression no common in English Language. It is indicate to change the sentence.

Line 15 to 16.     “are comprehensively investageted” is suggested only investigated

Line 18 to 22.    This sentence is hard to read, is suggested to change it

Line 23 to 24.    This sentence is hard to read, is suggested to change it

Line 23               “dinamic recristallization yelds” is a sentence not completely correct is suggested  to change it.

  • Introduction

Line 36                         Why Is reported this sentence of “dramatic enhanced” it is not perhaps required ?

Line 38             and so on, is require to indicate which are the field of interest that authors think are of priority. It is indicating to remove “and so on”.

Line 45-48         long sentence

Line 46                         ultra- strong. No clear this sentence

Line 58.                        It is difficult to understain this sentence

Line 101.          Formula required to be changed

Figura 1.          It Is difficult for the reader read the legend. It Is indicate to change them

Line 123           Extra space in reported in the text

  • Reference

Its indicated to check all references.

Author Response

 Thanks for these comments and suggestion.  Based on those detailed comments and suggestions, the manuscript was revised carefully, the quality of which was significantly improved.  All the changes made in the manuscript have been highlighted.  The “Response Letter” has a list of item-by-item response.  Please find it in the attachment.  Thanks. 

Reviewer 3 Report

Dear Authors,

your study presented valuable microstructural results about localized features of adiabatic shear bands (ASBs) in α+β dual-phase Titanium alloy.

The following points can maybe improve the quality of the work:

  • Lack of detailed information about mechanical test
  • Lack of information about the number of samples tested at the strain rate of 1.8×104 s-1
  • Why did not you perform the Test at other strain rates as well?

Author Response

Thanks for the positive recommendation and constructive suggestions.  The modifications in the manuscript have been highlighted in yellow.   The “Response Letter” has a list of item-by-item response.  Please find it in the attachment.  
